# Adherence Measures for Patients with Metastatic Castration-Resistant Prostate Cancer Treated with Abiraterone Acetate plus Prednisone: Results of a Prospective, Cluster-Randomized Trial

**DOI:** 10.3390/cancers12092550

**Published:** 2020-09-08

**Authors:** Henrik Suttmann, Jochen Gleissner, Andreas Huebner, Tim Mathes, Werner Baurecht, Katrin Krützfeldt, Hussein Sweiti, Susan Feyerabend

**Affiliations:** 1Private Practice, Urologikum Hamburg, 22399 Hamburg, Germany; suttmann@me.com; 2Private Practice, MVZ-DGU-Die GesundheitsUnion GmbH, 42103 Wuppertal, Germany; jgleissner@dgu-team.de; 3Private Practice, Center for Oncology and Urology, 18107 Rostock, Germany; wk-nord@hotmail.de; 4Institut für Forschung in der Operativen Medizin, University Witten/Herdecke, 51109 Cologne, Germany; Tim.Mathes@uni-wh.de; 5Biometrics, Acromion GmbH, 50226 Frechen, Germany; Werner.Baurecht@acromion-gmbh.com; 6Medical Affairs, Janssen-Cilag GmbH, 41470 Neuss, Germany; 7Clinical Development, Janssen Research and Development LLC, Pennsylvania, PA 19477, USA; HSweiti@its.jnj.com; 8Private Practice Studienpraxis Urologie, 72622 Nürtingen, Germany; praxis@studienurologie.de

**Keywords:** metastatic castration-resistant prostate cancer, abiraterone acetate, adherence, discontinuation, MMAS-4, quality of life

## Abstract

**Simple Summary:**

We report the final results of a multicenter, prospective, 2-arm trial in a real world setting for patients with metastatic castration resistant prostate cancer. A number of 675 patients were allocated by center-based cluster-randomization to arm A with adherence enhancing measures or arm B without adherence enhancing measures. Our study reveals a generally high medication adherence in patients with mCRPC with no clear difference between Arm A and Arm B. Our results confirm the benefit of Abiraterone acetate plus Prednisolone in terms of effectiveness and quality of life in a real world setting.

**Abstract:**

Residual androgen production causes tumor progression in metastatic, castration-resistant prostate cancer (mCRPC) patients. Abiraterone acetate (AA), a prodrug of abiraterone, is an oral CYP-17 inhibitor that blocks androgen production. It was hypothesized that adherence-enhancing measures (AEM) might be beneficial for mCRPC patients receiving abiraterone acetate plus prednisone (AA + P). This multicenter, prospective, 2-arm trial allocated mCRPC patients who were progressive after docetaxel-based chemotherapy or asymptomatic/mildly symptomatic after failure of an androgen deprivation therapy to Arm A (with AEM) or Arm B (without AEM) by center-based cluster-randomization. The primary objective was to assess the influence of AEM on discontinuation rates and medication adherence in mCRPC patients treated with AA + P. A total of 360 patients were allocated to Arm A, and 315 patients to Arm B. At month 3, the rate of treatment discontinuation, not due to disease progression or the start of new cancer therapy, was low in both arms (A: 9.0% vs. B: 7.3%, OR = 1.230). Few patients had a medium/low Morisky Medication Adherence Scale (MMAS-4) score (A: 6.4% vs. B: 9.1%, OR = 0.685). The results obtained did not suggest any clear adherence difference between Arm A and Arm B. In patients with mCRPC taking AA + P medication, adherence seemed to be generally high.

## 1. Introduction

Residual androgen production is a major cause of tumor progression in patients with metastatic castration-resistant prostate cancer (mCRPC). The combination of the oral CYP-17 inhibitor abiraterone acetate plus prednisone (AA + P) has been proven to be an effective mCRPC therapy in randomized controlled trials (RCT) [1,2,3,4]. However, little is known about the effectiveness of AA + P taken under routine conditions.

The effectiveness of self-administered therapies in real-world settings notably depends on medication adherence [5,6,7], defined as “the extent to which a patient acts in accordance with the prescribed interval and dose of a dosing regimen” [8]. Adherence in patients taking oral anti-cancer agents is often low [9], but adherence-enhancing measures (AEM) may have the potential to counteract this phenomenon [10]. Risk factors of non-adherence, including high age, comorbidities, and living alone [11,12], are common among patients with prostate cancer. Therefore, we hypothesized that AEM might be beneficial for mCRPC patients on AA + P.

The objectives of this cluster-randomized trial (CRT) were to assess the influence of AEM on discontinuation rates and medication adherence and to evaluate health-related quality of life (hr-QoL), fatigue, and survival in mCRPC patients treated with AA + P.

## 2. Methods and Materials

### 2.1. Trial Design

This multicenter, prospective, 2-arm CRT was conducted in accordance with the Declaration of Helsinki at 87 German urology or oncology ambulatory care facilities from October 2013 to June 2018. Patients were allocated to Arm A (with AEM) or Arm B (without AEM) by center-based cluster-randomization.

The trial was registered at the German Federal Institution for Drugs and Medical Devices as an observational study under the number 284 and approved by the ethics committee of the Medical Association Hamburg under the number PV4247 on 3 January 2013. The study protocol was amended twice and is available in Appendix A.

### 2.2. Patients

Adult men diagnosed with mCRPC who were either (i) progressive after docetaxel-based chemotherapy or (ii) asymptomatic/mildly symptomatic after failure of an androgen deprivation therapy were eligible for inclusion. The decision to initiate AA + P was made for all patients by the treating physician prior to and independent of this trial. All patients provided written informed consent.

### 2.3. Adherence-Enhancing Measures

To select AEM suitable for mCRPC patients, we reviewed AEM for patients taking oral anti-cancer agents [10,13]. Since non-adherence can be intentional (conscious decision not to take medication) or unintentional [14], we developed a multicomponent program with educational/counseling measures as well as reminder elements (Table 1). Some AEMs were optional to tailor the program to individual needs.

### 2.4. Endpoints

The primary endpoint was the rate of therapy discontinuation after 3 months for reasons other than disease progression, death, or the start of new cancer therapy (in the following “discontinuation”) [15]. The secondary endpoints were discontinuation after 6 months, reasons for and time to discontinuation, overall survival (OS), as well as the change of (i) self-reported medication adherence measured with the Morisky Medication Adherence Scale (MMAS-4; permission to use was granted by Prof. DE Morisky, MMAS Research, LLC, 2020 Glencoe Ave, Venice, CA 90291-4007, dmorisky@gmail.com) [16], (ii) hr-QoL measured with the Functional Assessment of Cancer Therapy-Prostate (FACT-P) questionnaire [17], and (iii) fatigue measured with the brief fatigue inventory (BFI) questionnaire [18]. In addition, physicians assessed the AEM.

Safety was monitored by collecting information on adverse events (AEs), clinical laboratory values, vital signs, and body weight. Additionally, tolerability was assessed by the physician. Version 17.0 of the Medical Dictionary for Regulatory Activities (MedDRA) was used for coding of AEs.

### 2.5. Cluster Randomization and Blinding

We allocated the clusters (ambulatory care facilities) in an equal ratio to the trial arms using a computer-generated randomization plan. All consecutive patients who fulfilled the eligibility criteria had to be documented at the participating study centers. At the time of center inclusion, the allocation was not known (allocation concealment). Blinding was not performed.

### 2.6. Sample Size

The determination of sample size was based on the pivotal abiraterone study, in which a discontinuation rate of 18% was observed after 12 weeks [1]. We expected discontinuation under real-world conditions to be higher than in the RCT [19] and considered a difference of 10% in discontinuation rates between the arms as relevant. Consequently, a discontinuation rate of 28% in the control arm (without AEM) was anticipated. The mean cluster size was expected to be 10, and the intra-cluster-coefficient (ICC) as 0.02. The power was set to 0.8, and two-sided alpha to 0.05. Considering a dropout rate of 10%, these assumptions led to a required sample size of 390 patients per arm.

### 2.7. Statistical Analysis

We used the software system SAS 9.4 for statistical analyses, which were confirmatory for the primary endpoint and exploratory for secondary endpoints. Analyses included all clusters and patients as initially randomized (intention-to-treat). Missing data were not substituted (observed case analysis).

We calculated ICCs for discontinuation and MMAS-4, according to Fleiss and Cuzick [20], and analyzed discontinuation with generalized linear mixed models to account for clustering. Sensitivity analysis of the primary endpoint included only patients of Arm A using any AEM, and all patients of Arm B. Following general recommendations regarding negative ICCs, analyses of MMAS-4 were performed without cluster adjustment (naïve analyses with logistic regression) [21].

We analyzed FACT-P and BFI as a change from baseline and used Cox-proportional hazard regression models for the analysis of OS. The analysis of OS included no random-effect to account for clustering since the design effect was close to 1 due to the small mean number of patients per cluster [22]. The analysis of BFI was performed for the subgroup of patients enrolled after the implementation of Amendment II.

In addition, we performed a post hoc co-variate cluster analysis to test if the impact of the AEM on discontinuation and MMAS-4 differs depending on baseline MMAS-4 or baseline FACT-P.

## 3. Results

### 3.1. Participant Flow and Baseline Characteristics

Overall, 694 patients were screened, of which 19 were excluded as screening failure. Thus, 675 patients were enrolled and analyzed. Forty-seven study sites comprising 360 men were allocated to Arm A, and 40 study sites comprising 315 men were allocated to Arm B (Appendix A). The mean cluster size was 7.66 (±5.88) for Arm A and 7.88 (±4.92) for Arm B.

Baseline characteristics were well balanced (Table 2). Self-reported adherence (MMAS-4) at the baseline was high (Arm A: 74.4%; Arm B: 81.0%).

### 3.2. Use of Adherence-Enhancing Measures during Study

Of all patients in Arm 1, 213 (59.2%) reported using AEM regularly. Thirty-two patients (8.9%) reported irregular use, 23 patients (6.4%) occasional use, and 65 patients (18.1%) no use at all. For 27 patients (7.5%), the use of AEM was unknown.

Physicians assessed the AEM as “very useful” for 72 patients (20.0%), as “useful” for 205 patients (56.9%), and as “not useful” for 83 patients (23.1%).

#### Treatment Discontinuation and Medication Adherence

Discontinuation rates at month 3 and 6 were comparable between the two study arms. Analyses with and without adjustment for clustering revealed very similar results (Table 3 and Figure 1a). The odds ratios (OR) for the sensitivity analysis of the primary endpoint were 1.249 (naïve analysis) and 1.135 (cluster-adjusted). The time to treatment discontinuation is presented in Figure 1b. The median duration of treatment was 310 days without a relevant difference between the two arms. The main reason for discontinuation other than disease progression or switch of therapy at month 6 was intolerance of therapy (Arm A: 18 (5.0%); Arm B: 11 (3.5%)).

At 3 and 6 months, Arm A comprised slightly more patients with high MMAS-4 score compared to Arm B (Figure 1c), but analyses of patients with medium/low MMAS-4 score revealed 95% CIs that included an OR of 1 (Table 3).

### 3.3. Health-Related Quality of Life, Fatigue, and Survival

FACT-P total scores were slightly increased at 3 and 6 months (Table 4). Mean change from baseline was comparable in both arms (3 months: 3.3 vs. 3.7; 6 months: 4.0 vs. 3.2). Overall, improvement of hr-QoL was observable for 128 patients (29.9%) at month 3 and for 110 patients (32.4%) at month 6.

The mean BFI total score did not change over time in any arm. Improvement of fatigue intensity and interference was observable for some patients (Table 4).

During the whole documentation period, 329 patients (48.7%) died. We found no difference in survival rates after 24 months (hazard ratio (HR) = 0.99; 95% CI: 0.80 to 1.23). The median survival time was 18.87 months.

### 3.4. Co-Variate Analyses

Medium/low baseline MMAS-4 increased the risk of discontinuation (OR = 2.52, 95% CI = 1.27 to 5.00) and medium/low MMAS-4 at 3 months (OR = 8.00; 95% CI = 3.62 to 17.54). Nevertheless, co-variate analyses did not indicate that the effect of the AEM on discontinuation (*p* = 0.6347) and MMAS-4 (*p* = 0.1408) at 3 months was modified by baseline MMAS-4. Likewise, we found no indication that baseline FACT-P modified the effect of the AEM regarding discontinuation (*p* = 0.2864) and MMAS-4 (*p* = 0.2172) at 3 months.

### 3.5. Safety

During the treatment period, 1928 adverse events (AEs) were documented for 505 patients (74.8%). The most commonly reported AEs mapped to MedDRA terms “fatigue”, “back pain”, and “urinary tract infection” (Table 5). AEs leading to permanent discontinuation of abiraterone occurred in 124 patients (18.4%), whereas 63 patients (9.3%) stopped treatment with prednisone due to an AE. Overall, 248 patients (36.7%) suffered a serious AE (SAE), with 76 SAEs leading to the death of 74 patients (11.0%).

Analyses of laboratory parameters, vital signs, and body weight did not suggest any clinically meaningful changes over time.

Physicians assessed tolerability of abiraterone mainly as good (for 351 patients, 52.0%) or very good (for 230 patients, 34.1%).

## 4. Discussion

The primary objective of the trial was to assess the influence of AEM on discontinuation rates in mCRPC patients treated with AA + P. Despite the use of multicomponent tailorable AEM, the rate of treatment discontinuation at 3 months was similar for patients using AEM and in the control arm. Likewise, we could not find a clear difference in self-reported medication adherence, although the OR suggested a slower drop in adherence for patients using AEM.

There are several explanations for the limited impact of our AEM. Firstly, discontinuation rates were generally low (overall 8.3%), with little potential for further improvement. Moreover, medication adherence was high, although mCRPC patients are known to have risk factors of non-adherence, and the intake of two drugs can be challenging. Previous studies in prostate cancer patients taking anti-androgens have found comparable high adherence rates, suggesting that non-adherence is not a general problem in men with prostate cancer [23]. Finally, AEM may not be a suitable approach to promote adherence for some patients, as indicated by the high rate (40.8%) of patients that did not utilize AEM regularly.

Our population of mCRPC patients was quite old, multi-morbid, and in an advanced disease stage. Thus, the generalizability of our findings to patients with better general health status and earlier disease stages may be limited. However, real-world studies from other countries on less co-morbid mCRPC patients being at an early disease stage have shown comparable low discontinuation and non-adherence rates [7,24,25]. Our specific AEM probably does not limit the generalizability of the results [26] since findings from other studies on the effectiveness of AEM for patients taking oral-anti-cancer agents are similar across different countries and settings: recently published systematic reviews have found little to no impact for most educational AEM as well as reminder components [10,12,27]. Additionally, a recently terminated RCT on a mobile-app-based individually tailored multicomponent AEM could not prove an effect on medication adherence in patients taking oral chemotherapy for various cancer types [28]. We found no difference in changes of FACT-P, BFI scores, or in OS between the two study arms. Overall, hr-QoL was slightly increased, and fatigue was stable in most patients at 3 and 6 months, which was in concordance with RCTs showing that treatment with AA + P significantly delayed hr-QoL deterioration [3]. The median survival time of 18.87 months was comparable to that of the pivotal COU-AA-301 trial, in which the median OS for post-chemotherapy mCRPC patients treated with abiraterone was 15.8 months [2].

The median treatment duration of 10 months was in line with the pivotal trials COU-AA-301 (8 months [1]) and COU-AA-302 (14 months [3]) and a recently published real-world retrospective cohort study (10 months [7]).

The incidence of AEs, SAEs, abnormal laboratory values/vital signs was also consistent with those reported in other trials. As previously published, the most common AEs were fatigue and back pain [1,3]. No new or unexpected safety findings were observed for AA + P.

Our study has some limitations that should be noted. Firstly, we did not reach the target sample size. In addition, we randomized clusters instead of patients because the implementation of the AEM on an individual level would have affected routine care and increased risk of bias as a result of physicians treating patients of both arms. Moreover, we used self-reports to measure medication adherence, which may tend to overestimate adherence [29]. Thus, we may have underestimated the impact of the AEM. Finally, the high amount of missing data for MMAS-4 and FACT-P and the late introduction of the BFI may also have introduced bias.

## 5. Conclusions

In patients with mCRPC taking AA + P, the risk of early therapy discontinuation due to other reasons than disease progression, death, or starting a new cancer therapy seemed to be low, and medication adherence seemed to be generally high. Our AEM could not further reduce discontinuation rates or increase overall medication adherence. However, our results indicated that AEM might potentially have a positive short-term effect on patients with a propensity to non-adherence. Thus, future studies on AEM in men with CRPC should be targeted towards patients that report difficulties with taking medication or proven non-adherence [30].

## Figures and Tables

**Figure 1 cancers-12-02550-f001:**
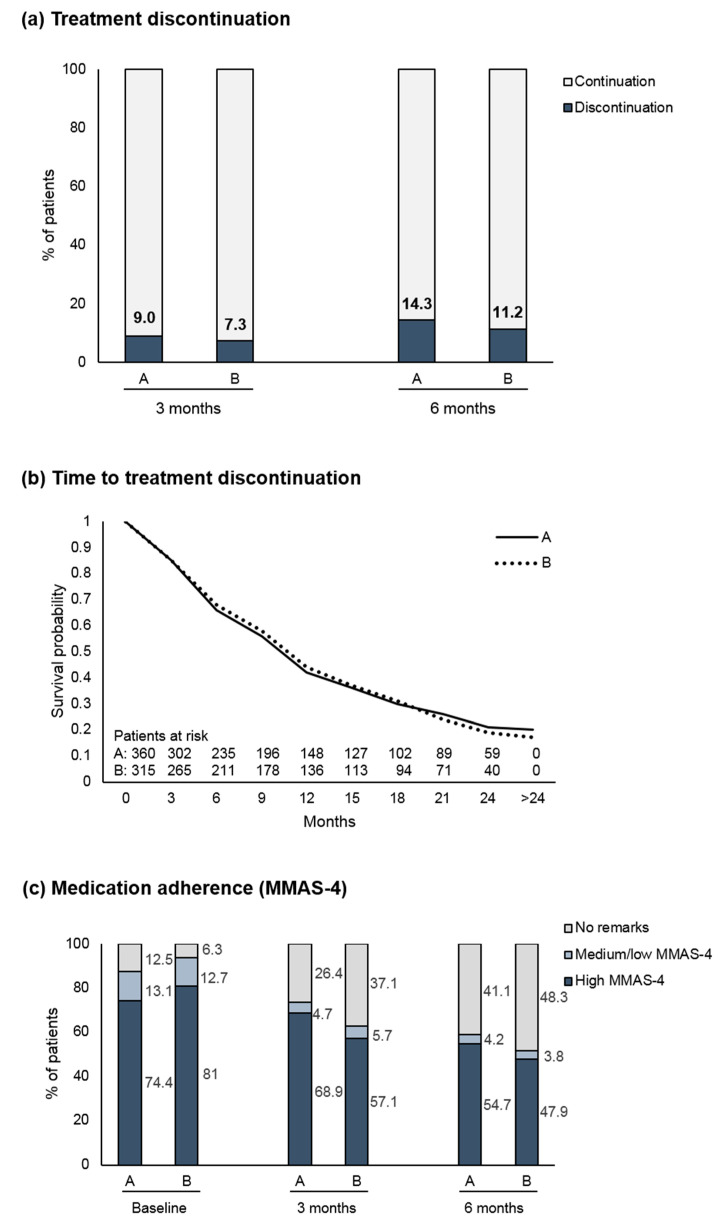
(**a**) Rates of treatment discontinuation after 3 and 6 months. Displayed are the discontinuations for reasons other than disease progression, death, or start of new cancer therapy, (**b**) Time to treatment discontinuation (Kaplan–Meier analysis), (**c**) Morisky Medication Adherence Scale (MMAS-4); Arm A: adherence-enhancing measures comprising educational/counseling measures as well as reminder elements, Arm B: no adherence-enhancing measures.

**Table 1 cancers-12-02550-t001:** Overview of the different adherence-enhancing measures.

Component	Type	Content (Frequency and Duration)	Intensity and Timing
**Mandatory**			
Educational video	Education	10-min video addressing mechanism of action, effectiveness, correct intake, and adverse events of AA + P	At first visit; watching could be repeated
Calls by a study nurse	Counseling and reminder	Structured interviews to identify problems with medication (e.g., side effects), intake (e.g., swallowing), and unintentional non-adherence (forgetting intake). In case of difficulties, possible solutions were discussed	During the first 3 months, every 2 weeks alternating with study visits, afterward, monthly in alteration with study visits
**Optional**			
Patient diary	Counseling and reminder	Monitoring intake and discussion with treating physician	na
Dosage card	Education and reminder	Planning of medication intake	na
Reminder SMS service	Reminder	Depending on intake schedule *	na

* abandoned after the inclusion of 200 patients due to the lack of interest by patients; AA + P: abiraterone plus prednisone; na: not applicable.

**Table 2 cancers-12-02550-t002:** Baseline characteristics.

Characteristic	Arm A ITT (*n* = 360)/BFI Set (*n* = 93)			Arm B ITT (*n* = 315)/BFI Set (*n* = 101)		
	*n*	Mean (SD)	*n* (%)	*n*	Mean (SD)	*n* (%)
Age in years	360	75.3 (7.4)	-	315	74.1 (8.0)	
Social support (living together with at least one person)	321	-	268 (83.5)	260	-	224 (86.2)
Higher education (university, college, or professional academy)	224	-	93 (41.5)	170	-	66 (38.8)
Gleason score	291	7.9 (1.2)	-	275	7.9 (1.2)	-
Charlson comorbidity index of 0	318		128 (40.3)	260	-	115 (44.2)
Previous chemotherapy	360		102 (28.3)	315	-	101 (32.1)
PSA	327	185.2 (475.6)	-	292	197.6 (483.9)	-
FACT-P total score	332	107.3 (22.8)	-	298	108.5 (23.0)	-
MMAS-4 (medium/low)	315	-	47 (14.9)	295	-	40 (13.6)
BFI score	93	2.9 (2.3)	-	101	2.7 (2.0)	-

FACT-P: Functional Assessment of Cancer Therapy-Prostate; MMAS-4: Morisky Medication Adherence Scale (permission to use from Prof. D. E. Morisky, MMAS Research, LLC, 2020 Glencoe Ave, Venice, CA 90291-4007, dmorisky@gmail.com); *n*: number of patients with available data; n: number of patients; PSA: prostate-specific antigen; SD: standard deviation; BFI: brief fatigue inventory.

**Table 3 cancers-12-02550-t003:** Treatment discontinuation and medication adherence at 3 and 6 months.

Variable	Arm A (*n* = 360)			Arm B (*n* = 315)				Arm A vs. Arm B	
	*n*	Naïve *n* (%), 95% CI	Cluster *n* (%), 95% CI	*n*	Naïve *n* (%), 95% CI	Cluster *n* (%), 95% CI	ICC	Naïve OR, 95% CI, *p*-Value	Cluster OR, 95% CI, *p*-Value
**3 months**									
Discontinuation	360	33 (9.2), 6.4 to 12.6	33 (9.0), 6.2 to 12.8	315	23 (7.3), 4.7 to 10.8	23 (7.3), 4.5 to 11.6	0.015	1.281, 0.735 to 2.232, 0.3818	1.230, 0.664 to 2.278, 0.5093
MMAS-4 (medium/low)	265	17 (6.4), 3.8 to 10.1	*	198	18 (9.1), 5.5 to 14.0	*	−0.063	0.685, 0.344 to 1.367, 0.2834	*
**6 months**									
Discontinuation	360	52 (14.4), 11.0 to 18.5	52 (14.3), 10.7 to 18.9	315	36 (11.4), 8.1 to 15.5	36 (11.2), 7.6 to 16.2	0.023	1.308, 0.830 to 2.062, 0.2466	1.303, 0.782 to 2.171, 0.3094
MMAS-4 (medium/low)	212	15 (7.1), 4.0 to 11.4	*	163	12 (7.4), 3.9 to 12.5	*	−0.063	0.958, 0.436 to 2.107, 0.9153	*

* no cluster analysis due to negative ICC; CI: confidence interval; ICC: intra-class correlation coefficient according to Fleiss and Cuzick, 1979; *n*: number of patients with available data; n: number of patients; MMAS-4: Morisky Medication Adherence Scale (permission to use from Donald E. Morisky, ScD, ScM, MSPH, Professor, MMAS Research, LLC, 2020 Glencoe Ave, Venice, CA 90291-4007, dmorisky@gmail.com); OR: odds ratio.

**Table 4 cancers-12-02550-t004:** Health-related quality of life, fatigue. and survival.

Variable	Total ITT (*n* = 675)/BFI set (*n* = 194)		
	*n*	Mean (SD)	*n* (%)
**3 months**			
FACT-P; change from baseline	428	3.5 (16.8)	-
FACT-P; patients with MID for improvement *	428	-	128 (29.9)
BFI; change from baseline	146	−0.2 (1.8)	-
BFI; patients with MID for improvement of fatigue intensity **	146	-	31 (21.2)
BFI; patients with MID for improvement of fatigue interference ***	146	-	11 (7.5)
**6 months**			
FACT-P; change from baseline	339	3.6 (20.0)	-
FACT-P; patients with MID for improvement *	339	-	110 (32.4)
BFI; change from baseline	339	0 (2.0)	-
BFI; patients with MID for improvement of fatigue intensity **	122	-	26 (21.3)
BFI; patients with MID for improvement of fatigue interference ***	122	-	7 (5.7)
**24 months**			
	*n*	median (95% CI)	
Survival in months	675	18.87 (17.73 to 20.60)	

* increase of 10 points from baseline; ** baseline value ≥5 points and decrease by ≥2 points; *** baseline value ≥5 points decrease by ≥1.25 points; BFI: brief fatigue inventory; CI: confidence interval; FACT-P: Functional Assessment of Cancer Therapy-Prostate; ITT: intent-to-treat analysis set; MID: minimal important difference; *n*: number of patients with available data; n: number of patients; SD: standard deviation.

**Table 5 cancers-12-02550-t005:** Adverse events.

Subgroup	Total (*n* = 675)	
	*n* (%)	E
Patients at risk	675 (100.0)	-
Patients with at least 1 AE *(treatment period)	505 (74.8)	1928
Patients with abiraterone-related AEs **	204 (30.2)	340
Patients with prednisone -related AEs **	141 (20.9)	212
Patients with AEs leading to permanent stop of abiraterone	124 (18.4)	147
Patients with AEs leading to permanent stop of prednisone	63 (9.3)	70
Most common AEs by preferred term		
Fatigue	68 (10.1)	78
Back pain	64 (9.5)	71
Urinary tract infection	41 (6.1)	54
Patients with at least 1 SAE * (treatment period)	248 (36.7)	476
Patients with abiraterone-related SAEs **	30 (4.4)	35
Patients with prednisone-related SAEs **	18 (2.7)	20
Patients with SAEs leading to death	74 (11.0)	76
Patients with SAEs leading to permanent stop of abiraterone	72 (10.7)	81
Patients with SAEs leading to permanent stop of prednisone	63 (9.3)	70
Most common SAEs by preferred term		
General physical health deterioration	20 (3.0)	21
Death	18 (2.7)	18
Hematuria	12 (1.8)	15
Hydronephrosis	12 (1.8)	15
Pneumonia	12 (1.8)	12

* multiple responses possible; ** drug-related=definite, probable, possible; AE: adverse event; E: episodes (number of coded preferred terms); *n*: total number of patients; n: number of patients; SAE: serious adverse event.

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
