# Peer review of "Adherence Measures for Patients with Metastatic Castration-Resistant Prostate Cancer Treated with Abiraterone Acetate plus Prednisone: Results of a Prospective, Cluster-Randomized Trial"

_cancers, 2020, doi:10.3390/cancers12092550_

Round 1

Reviewer 1 Report

Novel concept to improve treatment outcomes for CRMPC

Results seem counter intuitive would expect better outcomes with AEM ? why this occurred

May be due to Health Care system in Germany

Author Response

Dear Reviewer, thank you for your review and comment. We also expected a better outcome with AEM. Some possible reasons why this was not the case have been mentioned in the manuscript. The assumption that it may be due to the good health care system in Germany might be correct as well since we observed that the adherence of these patients is already quite high without special adherence measures.

Reviewer 2 Report

Drs. Suttmann, Krutzfeldt and team have organized an interesting CRT clinical trial to determine AA+P medication adherence in 2 arms of mCRPC patients, one arm with adherence-enhancing measures (AEM) and one without.

The methods defined  AEM, endpoints, sample size (not achieved) and predicted discontinuation rates / power analysis. Describing these details in this manuscript is  important. The statistical analysis was also provided. The methods and discussion sections were well referenced for comparison to other studies. 

Results were  interesting in that 1) no clear adherence difference between arms was observed and 2) at 3 months the discontinuation rates for both arms were <10.0%. This is lower that what deBono/Logothetis observed in 2011 but similar to more recent studies. However, concerns about bias and that the designed AEP did not further reduce discontinuation in both arms was a concern for the authors. The authors suggestion to further define patient cohorts by selecting  men with issues  taking medication in general may help with the AEP.

This manuscript is interesting to select group of scientists and is well referenced. 

Author Response

Dear Reviewer, many thanks for your review. We agree with your comments.

Reviewer 3 Report

In this report, the authors developed adherence-enhancing measures (AEM) and aimed to assess whether AEM could influence the discontinuation rates in mCRPC patients who were treated with AA+P, and this trial is termed IMPACT. The results suggest that AEM could not increase the overall medication adherence, although may have some short-term benefits. And the authors discussed the potential limitations in the this study, such as smaller sample size, the route of reporting. Overall, the study is potentially very important for improving the life quality of mCRPC patients, but future studies need further adjustments and optimizations.

Specific comments:

1.The authors term their trial as IMPACT, while there is another ongoing clinical trial in prostate cancer also termed IMPACT(NCT00261456): Identification of Men With a Genetic Predisposition to ProstAte Cancer. It is better to change the name to avoid confusion.

2.It is not clear what is the difference between AEM and AEP, are they interchangeable?

3.There should be a comparison between arm A and B in Table 4.

4.As shown in Table 2, most of the patients have social support (living together with at least one person). It is suggested that the companion of these patients can also get engaged in the reporting process, which could potentially increase the accuracy.

Author Response

Dear Reviewer, many thanks for your review and comments.

Point 1: Since the acronym IMPACT was used for the study since the beginning and the study was submitted to the Ethics Committee and to the regulatory authority with this acronym we would not like to change it to another acronym at this stage. If you think it could be misleading the acronym can be deleted from the title of the manuscript.

Point 2: You are right that the terms AEM and AEP have been used in the manuscript interchangable. To avoid any confusion we have changed this in the attached revised manuscript and only use AEM.

Point 3: For the endpoints described in table 4 we have not seen any clear difference between Arm A and Arm B which is also described in the text of the manuscript. This is the reason why we decided to show these endpoints for the total study population in table 4 and not for both arms. I hope this is acceptable.

Point 4: Even though it was asked if patients are living together with at least one person the companion of the patient was not actively involved in the study and the adherence measures. We agree that this is a good suggestion for further research in this field.